# Arterial Hypertension in Morbid Obesity after Bariatric Surgery: Five Years of Follow-Up, a Before-And-After Study

**DOI:** 10.3390/ijerph19031575

**Published:** 2022-01-29

**Authors:** Angeles Arias, Cristobalina Rodríguez-Álvarez, Enrique González-Dávila, Alfonso Acosta-Torrecilla, M. Mercedes Novo-Muñoz, Natalia Rodríguez-Novo

**Affiliations:** 1Department of Preventive Medicine and Public Health, University of La Laguna, 38200 Santa Cruz de Tenerife, Spain; crrodrig@ull.edu.es; 2Department of Mathematics, Statistics and Operations Research, University of La Laguna, 38200 Santa Cruz de Tenerife, Spain; egonzale@ull.edu.es; 3Canary Islands Health Service, 38530 Santa Cruz de Tenerife, Spain; alfore1965@gmail.com; 4Faculty of Health Sciences, Nursing Section, University of La Laguna, 38200 Santa Cruz de Tenerife, Spain; mernov@ull.edu.es (M.M.N.-M.); nrodrigu@ull.edu.es (N.R.-N.)

**Keywords:** morbidly obese patients, bariatric surgery, arterial hypertension, weight loss

## Abstract

Background: Arterial hypertension (HTN) is common among morbidly obese patients undergoing bariatric surgery. The aim of this study is to analyse the prevalence and evolution of HTN and weight loss in patients suffering from morbid obesity before and after bariatric surgery, during a follow-up period of five years. Methods: A before-and-after study was carried out on severely obese patients undergoing Laparoscopic Roux-En-Y Gastric Bypass (LRYGB). Criteria for HTN diagnosis were current treatment with antihypertensive agents and/or systolic blood pressure (SBP) > 140 mmHg and/or diastolic (DBP) > 90 mmHg. HTN remission was defined as normalisation of blood pressure (BP) maintained after discontinuation of medical treatment, and HTN recurrence was considered when HTN diagnostic criteria reappeared after remission. Weight loss during the study period was evaluated for each patient, calculating excess weight loss percentage (% EWL) and BMI loss percentage (% BMIL) with reference to the baseline value. Results: A total of 273 patients were included in the study. HTN was present in 48.2%; 29.4% of hypertensive patients showed HTN remission two years after the surgical procedure, 30.3% of them had relapsed at five years. Conclusion: LRYGB in obese patients is associated with a remission of HTN, and no weight loss differences were observed between the group of patients showing HTN remission at two years and the group who did not. However, differences were observed after the second follow-up year, with an increased weight loss in the remission group, which could indicate that sustained weight loss favours the control of HTN.

## 1. Introduction

Epidemiological studies continue to show an increasing prevalence of obesity worldwide [1,2,3,4,5,6], which has been strongly associated with the development of HTN, along with other major metabolic and cardiovascular risk factors, such as dyslipidaemia and type II diabetes mellitus [7,8,9,10].

Raised blood pressure (BP), commonly defined as SBP ≥ 140 mmHg or DBP ≥ 90 mmHg, is used to identify individuals at high risk of cardiovascular diseases. One of the global non-communicable disease (NCD) targets adopted by the World Health Organisation (WHO) in 2013 is to reduce the prevalence of raised BP by 25%, compared with its 2010 level, by 2025 [11].

Dietary treatment, lifestyle changes, exercise, and behavioural therapy, together with treatment with complementary medication achieve weight loss, thus improving obesity-linked comorbidities, including HTN. However, these treatments do not provide the expected results in individuals suffering from morbid obesity, with different authors indicating that bariatric surgery is the most efficient procedure to control obesity and its comorbidities [9,11,12,13,14,15,16]. However, other authors report that some of the patients regain weight in the long term, which can worsen related comorbidities [11,17,18,19,20,21].

HTN in obesity is a result of a complex relationship of multiple neurohormonal factors. Weight loss remains a first-line therapy in the treatment of HTN [9], and, in the case of morbid obesity, several studies state that the effect on HTN is often observed soon after surgery, although long term follow-up data are limited and suggest this effect decreases overtime [9,14].

The aim of this study is to analyse the prevalence and evolution of HTN and weight loss in patients suffering from morbid obesity, before and after bariatric surgery, during a follow-up period of five years, as well as studying the differences in comorbidities between patients who maintained HTN at two years and those who did not, as well as the rate of relapse in patients at two and five years of follow-up.

## 2. Materials and Method

### 2.1. Study Design

A before-and-after study was conducted on a consecutive, non-randomised sample, including all patients suffering from morbid obesity submitted to bariatric surgery in the Surgery Service of the Public Reference Hospital between January 2010 and December 2014. The number of patients submitted to surgery was 273 (69 men and 204 women), aged 20 to 61 and 19 to 64, respectively. Exclusion criteria were patients with incomplete or missing data at any time point during follow-up (due to change of address, death, or other causes).

According to the study protocol, all patients were evaluated in their pre- and post-operative phases for a period of five years after surgery.

BP was taken using an Omron M6 AC automatic digital monitor (Omron Healthcare, Kyoto, Japan). Three BP readings were taken at one-minute intervals, and the mean value of the second and third readings was used for study analysis. BP was measured in the seated position for ≥5 min in a quiet room, with the bladder empty, and the arm at heart level. HTN was considered when patients either presented SBP ≥ 140 mmHg or DBP ≥ 90 mmHg, or were taking antihypertensive medication [22].

A five-year post-surgery follow-up was performed with patients identified as suffering from HTN. Remission was considered when the criteria applied were not met. HTN remission was defined as normal BP levels without antihypertensive therapy during the study period, and, finally, HTN recurrence was considered when HTN diagnostic criteria reappeared after remission. Polymedicated patients were defined as those who took three or more medications.

Type 2 diabetes was considered when glucose ≥ 126 mg/dL or HbA1c > 7.0 mmol/L, and dyslipidaemia was considered when total cholesterol > 200 and/or HDL cholesterol < 40 men < 50 women and/or triglycerides > 200 and/or use of therapy.

Demographic, as well as anthropometric, biochemical, and clinical analyses were included. BMI is defined as the individual’s body weight divided by the square of his/her height, excess weight loss (EWL) as the difference between the current weight and the weight estimated for a BMI of 25 kg/m^2^, and excess of BMI loss (BMIL) as the difference between the current BMI and the BMI of 25 kg/m^2^.

Weight loss during the study period was evaluated for each patient, calculating the percentage of excess weight loss (% EWL) and the percentage of BMI loss regarding the baseline value (% BMIL).

Finally, an analysis of the variables of weight, % EWL, and % BMIL was carried out, comparing the patients who presented hypertension remission at two years with those who did not, as well as those who presented hypertension recurrence at five years with those who did not.

The surgical technique used was LRYGB, which is a mixed technique that combines restrictive and malabsorptive procedures. Its main features are the creation of a gastric “pouch” (25 ± 5 mL), an alimentary loop of 100 cm, a biliopancreatic loop of 60 cm, and a common channel. Surgeries were performed by the same group with a standardisation of the processes and technique.

The study protocol was approved by the local Ethics Committee, and complies with the Declaration of Helsinki. According to the regulations of the Hospital Ethical Committee, no approval was required for this before-and-after study, which was registered in the Research Registry Platform, and reported according to the STROCSS criteria [23].

### 2.2. Statistical Analysis

Continuous variables were summarised with the mean ± standard deviation (SD), and categorical and discrete variables with frequency and percentage. The reduction or increase between periods is shown with a mean and 95% confidence interval (CI). Comparisons between two periods were performed with a paired *t*-test and McNemar chi-squared test, respectively. A repeated measure analysis of variances (RM-ANOVA) was applied to study the evolution of SBP, DBP, % EWL, and % BMIL in patients who initially presented HTN and those who did not, including sex as a covariable. Analyses were performed using SPSS 25 (IBM SPSS, Armonk, NY, USA), and differences with a *p*-value < 0.05 were considered statistically significant.

## 3. Results

Throughout the study period, a total of 273 patients underwent bariatric surgery. Of the 247 patients who could be monitored during the entire follow-up period, initially, 119 presented HTN (48.2%). Seventeen percent of the patients (42) required a second intervention. However, no significant differences were found (*p*-value = 0.398) between the group with hypertension problems, 19.3%, and the normotensive group at the beginning of the study, 14.8%.

The characteristics of patients with and without HTN are shown in Table 1. The mean age of the cohort was 39.1 ± 9.7. A high percentage suffered from diabetes, dyslipidaemia, and HTN. It is observed that hypertensive patients are older and have a higher BMI, as well as presenting more diabetes, dyslipidaemia, comorbidities, and polymedication than normotensives, with significant differences between the two groups, and no differences regarding sex or smoking habit.

Figure 1 shows the evolution of HTN (systolic and diastolic) and weight loss expressed as % EWL and % BMIL during the follow-up period, distinguishing between hypertensives and normotensives.

The group of hypertensive patients shows a significant decrease, *p* < 0.001, of SBP at two years, −12.2 mmHg 95% CI (−15.6; −8.7), and a subsequent slight increase at five years, 1.1 mmHg 95% CI (−1.9; 4.1), *p* = 0.475. In the normotensive group, SBP remains more stable in the follow-up period, although a significant increase is observed at two years, 3.7 mmHg 95% CI (2.3; 5.1), *p* < 0.001, which subsequently decreases at five years, so that the increase is not significant regarding the baseline value (1.3 mmHg 95% CI (−0.6; 3.1), *p* = 0.174). There are no significant differences in sex (*p* = 0.744) or in the difference in evolution over time (*p* = 0.617, interaction time * sex).

A significant reduction of DBP is observed in the hypertensive group at two years, −7.3 mmHg 95% CI (−9.7; −5.0), *p* < 0.001; the reduction between two and five years is not significant, −1.3 mmHg 95% CI (−3.5; 0.9), *p* = 0.251. However, in the normotensive group, DBP remains stable during the two first years (*p* = 0.299), significantly decreasing between years two and five, −2.7 mmHg 95% CI (−4.8; −0.6), *p* = 0.012. No significant differences are observed in evolution between the hypertensive and normotensive groups from two to five years (*p* = 0.052, interaction effect).

At the beginning of the study, 30.2% of the patients were polymedicated, whereas at five years, this was 21.2% (*p*-value < 0.001). Of the group that were polymedicated at the beginning, only 48.6% continued to be so after five years, whereas in the group that were not polymedicated at the beginning of the study, only 9.4% were polymedicated at five years.

In the hypertensive group, % EWL significantly differs (*p* < 0.001) when comparing each period with its preceding period, observing a maximum % EWL at two years, 66.2 ± 19.4%, which decreases, however, at four and five years, reaching 60.3 ± 24.9% in the latter case. Although % EWL remains stable within the hypertensive and normotensive groups during the first phase, after the second year, this weight loss is higher in the normotensive group (*p*-value < 0.004 at 2, 4 and 5 years). Significant differences between consecutive periods are also observed in the normotensive group (*p* < 0.001), reaching their peak at four years, with a percentage of 74.7 ± 24.9, and then experiencing a considerable decrease to 70.1 ± 27.4% at five years, with a similar behaviour to that observed in the hypertensive group (*p* = 0.107, interaction effect).

As for % BMIL, its behaviour is similar to that observed in the % EWL, although significant differences between hypertensives and normotensives (*p* = 0.047) are only observed at four years. Maximum % BMIL was 31.3 ± 9.6 in hypertensives at two years in contrast to 33.0 ± 10.7% found in the normotensive group at four years.

Among the 119 patients suffering from initial HTN, 71 (60%) continued to be hypertensive both at two and five years, and 36 patients (30%) were in remission at 5 years (23 of them from 2 years). HTN remitted at 2 years in 12 patients (10%), but they were hypertensive at 5 years.

However, of the 128 normotensive patients, 20 (15.6%) had HTN at 2 years. The characteristics of hypertensive patients, with or without remission, at 2 years are shown in Table 2. Significant differences were found between the variables SBP and diabetes mellitus.

Figure 2 shows that % EWL and % BMIL behave similarly in the group that experienced remission and in the group who did not, displaying a trend reversal close to being significant between years four and five, with *p*-values of 0.073 and 0.056 (interaction effect). The group that did not experience remission had the highest values of % EWL and % BMIL at two years (65.7 ± 20.5% and 31.0 ± 10.1%, respectively), although there is a decrease in these percentages from then on. The mean falls of % EWL between years two and five were 7.6% (95% CI 2.7; 12.6, *p*-value = 0.003) and 3.7% (95% CI 1.4; 6.0, *p*-value = 0.002) in % BMIL. It can be clearly seen that the group that experienced remission maintains the maximum values found at two years (67.4 ± 16.9% and 31.9 ± 8.1%, respectively) during the rest of the follow-up period, with a fall that varied between 1.7% in % EWL (95% CI −4.5; 7.9, *p*-value = 0.573) and 0.6% in % BMIL (95% CI −2.3; 3.7, *p*-value = 0.659).

Table 3 shows the results of the HTN remission prediction using logistic regression. It is observed that the probability of remission increases with lower age, initial SBP, and total cholesterol, as well as with higher hyperuricemia and diabetes.

Recurrence is considered when HTN reappeared after remission. Of the thirty-five hypertensive patients in remission at two years, twelve (34.3%) had relapsed at five years. Table 4 shows that the trend for recurrence occurs in older patients, those with dyslipidaemia, and smokers or ex-smokers, but without significant differences.

If a multivariate analysis is applied (logistic regression) on recurrence, only HDL cholesterol would remain in the model, but with a *p*-value of 0.084, indicating that it occurs in patients with a tendency to have a lower initial HDL cholesterol. If this variable was suppressed, the variable remaining in the model would be dyslipidaemia (%), but also with a *p*-value over 0.05.

Figure 3 shows non-significant differences in the % EWL or % BMIL values between recurrent and non-recurrent patients at five years who presented HTN remission at two years. Regarding % BMIL, coincidence is almost perfect (*p*-value = 0.814), unlike the % EWL, although without significant differences (*p*-value = 0.385). Recurrent patients generally present slightly lower values than non-recurrent patients.

## 4. Discussion

This study presents results on arterial HTN and weight loss in patients after bariatric surgery using LRYGB, with a follow-up period of five years. Before surgery, the prevalence of hypertension among all patients who attended the bariatric surgery unit was 48.2%, similar to that obtained by Benaiges et al. [16] and Hawkins et al. [24], and lower than that reported by other authors [25,26,27]. Cadena-Obando [27] considered presenting arterial HTN as a risk factor contributing to weight gain two years after bariatric surgery.

Hypertensive patients in the present study are older, have higher BMI, and present increased values of blood glucose, HbA1C, and total cholesterol, while having lower values of HDL cholesterol, thus suffering more from dyslipidaemia and diabetes than normotensive patients. All the studies reviewed in the literature state that morbidly obese individuals tend to have a high number of comorbidities which, in addition, increase with age and BMI [18,27,28].

The mean age of the cohort is 36.6 ± 9.3 in normotensive patients, and is higher in hypertensive ones (41.7 ± 9.5), similar to that reported by Hawkins et al. [24] and Benaiges et al. [16].

The levels of SBP and DBP in hypertensive individuals decreased in the first two years, although SBP increased significantly between years two and five after surgery, whereas DBP continued decreasing considerably throughout the study period. As regards normotensive individuals, both SBP and DBP increased slightly in the first two years, and then started decreasing progressively until 5 years after surgery. Benaiges et al. [16] report that, in both groups, SBP and DBP decreased during the first six months after surgery, at which point SBP started increasing slightly, but progressively, during the follow-up period, whereas DBP remained stable, although their follow-up period was three years, and the period was five years in the present study.

In the present study, weight loss comparison between hypertensive and normotensive individuals after bariatric surgery using % EWL and %BMI values shows both groups evolve in a similar way during the first phase, where weight decreases, but after two years, initially hypertensive individuals tend to present smaller values of weight loss than normotensives. Thus, hypertensives show a tendency to reduce % EWL and % BMIL values after two years, whereas in normotensive individuals, this reduction occurs at four years.

Various authors report HTN decreases soon after surgery, and evolves independently from weight loss as, in many cases, weight continues to decrease during the follow-up years, whereas BP stabilises or increases [9,25,29].

The percentage of HTN remission in the study here two years after surgery was 29.4%, which is lower than that found by other authors, although this is difficult to compare due to the different follow-up periods. Neff et al. [26] found a remission percentage of 32% one year after laparoscopic surgery, and 23% at five years. In other words, it decreased with time. They observed that those with remission of hypertension (remitters) had greater total % EWL (37 ± 1% vs. 27 ± 1%, *p* < 0.01) than those without remission (non-remitters). At five years, this difference persisted, with remitters having greater total % EWL (27 ± 2% vs. 23 ± 1%, *p* = 0.02) than non-remitters. The GATEWAY randomised trial by Schiavon et al. [30] indicated that, a year after bariatric surgery, 51% of the patients submitted to gastric bypass showed remission of hypertension, a value which is equal to that previously obtained by Shah et al. [31], with a follow-up of five years. Jakobsen et al. [15] reported a remission percentage of 31.9% with a mean follow-up of six-and-a-half years after surgery, indicating this remission had been higher than that of those who underwent non-surgical specialised medical treatment whose remission percentage was 12.4%.

Benaigues et al. [16] report a remission rate of 68.1% a year after bariatric surgery, which is higher than the other studies mentioned. They also found a higher remission rate in cases of greater weight loss, measured as % EWL and % BMIL. In the present study, % EWL and % BMIL remained more stable in the remitters group; although, a tendency to decrease was observed at two years in the non-remitters group.

It is observed that patients who normalise their BP two years after surgery tend to be younger, with lower initial values of SBP, higher hyperuricemia, and lower HDL cholesterol, and there is a higher percentage of diabetics (glucose also tends to be higher) due to the fact that, within the remitters group, 74% are diabetics, whereas in the non-remitters group, only 52% are diabetic.

In the present study, out of the 128 patients who did not present HTN initially, 20 presented it at 2 years (15.6%), which is considerably higher than the study by Benaiges et al. [16], who reported that only 1 normotensive patient developed HTN during the follow-up period, and this comorbidity improved in 23% of them, whereas only 9.6% did not improve.

When analysing those who show remission at two years and recurrence at five years, one can see that, in general, there is a higher probability of recurrence within older individuals, and those who smoke more, have lower HDL cholesterol, and suffer from dyslipidaemia. The remission rate at five years was 30% in the study here, whereas that of Benaiges et al. [16] was 53.2% at three years, though they are not comparable due to the different follow-up periods, and they find a correlation between weight loss and recurrence; thus, recurring individuals present lower weight loss values. However, this correlation was not significant in the present study. No significant differences were found between the characteristics of patients presenting recurrence at five years and those who do not, probably due to the relatively low number of recurring patients. Nevertheless, the tendency is to recur in older individuals, and former or current smokers, and in those presenting lower initial HDL cholesterol, higher triglycerides, and/or dyslipidaemia.

### Limitations of the Study

In this type of study, with a long follow-up period, the accuracy of the measurement of comorbidities can be affected by measurement variability. Moreover, various factors, such as diet, physical exercise, salt intake, among others, which are known to have an impact on HTN remission or relapse were not included in the study. Other aspects, such as BP measurement over twenty-four hours, which would give more reliable measurements, or heart function parameters, were not studied. Another limitation of the study is that laboratory data or biomarkers were not determined to illustrate the alterations of hypertension, including remission and recurrence.

However, considering the length of the follow-up period, and the fact that it was conducted with a high percentage of the initial cohort, with the percentage of loss being 6.2%, the authors believe the study is of much interest to assess the improvement of arterial HTN.

## 5. Conclusions

A third of the patients in the present study showed remission of HTN at five years of follow-up after bariatric surgery. There were differences after the second follow-up year, with the remitters group showing greater weight loss, which could suggest that maintained weight loss favours the control of arterial HTN.

## Figures and Tables

**Figure 1 ijerph-19-01575-f001:**
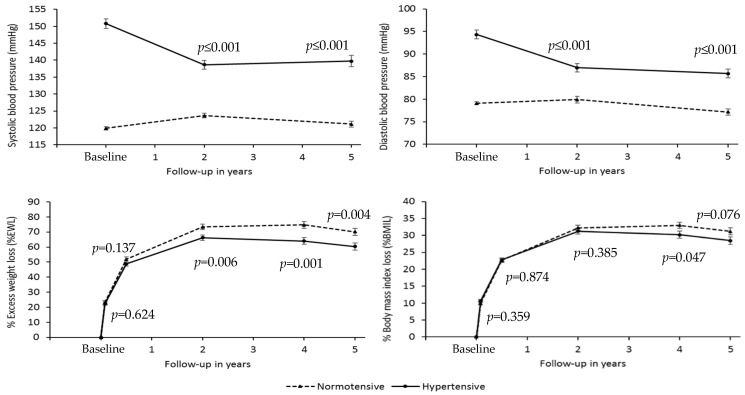
Changes in blood pressure, percentage of excess weight loss, and percentage of body mass index loss during the follow-up period in hypertensive and normotensive patients. The *p*-values in the figure correspond to the differences between groups in the different periods. The bars show a standard error. A repeated measures model has been applied.

**Figure 2 ijerph-19-01575-f002:**
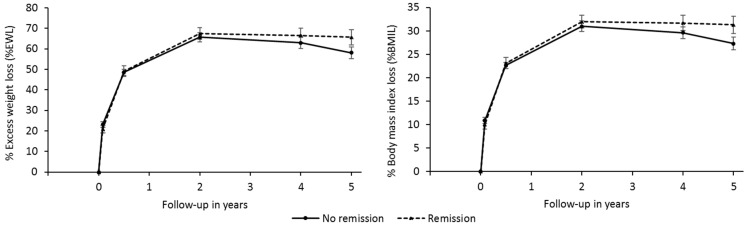
Comparison of weight loss with remission over the five-year period.

**Figure 3 ijerph-19-01575-f003:**
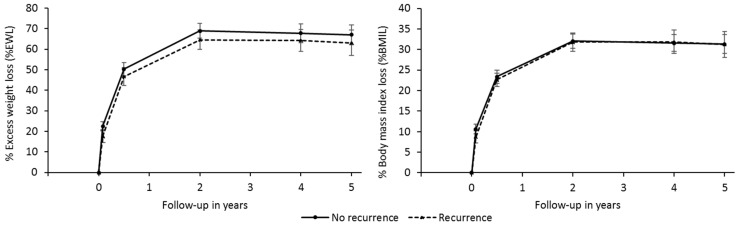
Relationship between recurrence over the five-year period and weight loss (% EWL and% BMIL).

**Table 1 ijerph-19-01575-t001:** Baseline characteristics of normotensive and hypertensive patients.

	Normotensive	Hypertensive	*p*-Value	Total
	(*n* = 128)	(*n* = 119)		(*n* = 247)
Age (years)	36.6 ± 9.3	41.7 ± 9.5	<0.001	39.1 ± 9.7
Women (%)	100 (78%)	87 (73%)	0.376	187 (76%)
Smoker (%)			0.067	
Current	21 (16%)	16 (13%)		37 (15%)
Ex	2 (2%)	9 (8%)		11 (5%)
Body mass index (kg/m^2^)	46.7 ± 7.9	48.8 ± 7.8	0.040	47.7 ± 7.9
Systolic BP (mmHg)	120 ± 5.4	151 ± 15.7	<0.001	135 ± 19.3
Diastolic BP (mmHg)	79 ± 4.1	94 ± 10.9	<0.001	86 ± 11.1
Glucose (mg/dL)	102 ± 30.9	127 ± 50.7	<0.001	114 ± 43.4
HbA1C (%)	7.1 ± 0.6	7.5 ± 1.0	0.011	7.4 ± 0.9
Hyperuricemia (mg/dL)	5.2 ± 1.3	5.3 ± 1.3	0.496	5.3 ± 1.3
Total Cholesterol (mg/dL)	186 ± 40.5	199 ± 40.9	0.013	192 ± 41.2
LDL Cholesterol (mg/dL)	119 ± 29.8	123 ± 33.7	0.398	121 ± 31.7
HDL Cholesterol (mg/dL)	49 ± 10.8	46 ± 9.8	0.043	47 ± 10.4
Triglycerides (mg/dL) *	122 (99; 149)	128 (102; 160)	0.170	125 (101; 156)
Dyslipidaemia (%)	86 (67%)	94 (79%)	0.045	180 (73%)
Type 2 diabetes mellitus (%)	38 (30%)	70 (59%)	<0.001	108 (44%)
Polymedicated patients	16 (13%)	58 (49%)	<0.001	74 (30%)
Number of comorbidities				
0–2	70 (54.6%)	31 (26.6%)	<0.001	
3–5	54 (42.3%)	49 (41.1%)		
>5	4 (3.1%)	38 (32.8%)		

Data are given as mean ± SD or percentage (%), except: * median (P25; P75).

**Table 2 ijerph-19-01575-t002:** Baseline clinical characteristics of patients with and without hypertension remission two years after bariatric surgery.

	Hypertension Remission (*n* = 35)	No Hypertension Remission (*n* = 84)	*p*-Value
Age (years)	39.4 ± 9.6	42.6 ± 9.4	0.090
Women (%)	22 (63%)	65 (77%)	0.117
Smoker (%)			0.187
Current	5 (14%)	11 (13%)	
Ex	5 (14%)	4 (5%)	
Body mass index (kg/m^2^)	49.0 ± 7.4	48.7 ± 8.0	0.879
Systolic BP (mmHg)	146 ± 15.6	153 ± 15.3	0.019
Diastolic BP (mmHg)	94 ± 9.0	94 ± 11.7	0.745
Glucose (mg/dl)	141 ± 57.9	122 ± 46.6	0.056
HbA1C (%)	7.4 ± 0.8	7.6 ± 1.1	0.365
Hyperuricemia (mg/dL)	5.7 ± 1.3	5.2 ± 1.2	0.053
Total Cholesterol (mg/dL)	189 ± 44.2	203 ± 39.0	0.090
LDL Cholesterol (mg/dL)	119 ± 36.0	125 ± 32.8	0.401
HDL Cholesterol (mg/dL)	43 ± 9.2	47 ± 9.8	0.051
Triglycerides (mg/dL) *	122 (110; 170)	133.5 (101; 160)	0.825
Dyslipidaemia (%)	25 (71%)	69 (82%)	0.191
Type 2 diabetes mellitus (%)	26 (74%)	44 (52%)	0.040

Data are given as mean ± SD or percentage (%), except: * median (P25; P75).

**Table 3 ijerph-19-01575-t003:** Logistic regression for prediction of remission at two years.

	Odds Ratio	95%CI	*p*-Value
Age	0.948	0.901–0.997	0.038
Systolic BP initial	0.954	0.923–0.986	0.006
Hyperuricemia	1.441	1.021–2.035	0.038
Total Cholesterol	0.989	0.978–1.001	0.077
Type 2 diabetes mellitus	4.171	1.526–11.401	0.005

**Table 4 ijerph-19-01575-t004:** Baseline clinical characteristics of patients with and without hypertension recurrence five years after bariatric surgery.

	Hypertension Recurrence (*n* = 12)	No Hypertension Recurrence (*n* = 23)	*p*-Value
Age (years)	41.5 ± 8.3	38.3 ± 10.2	0.352
Women (%)	8 (67%)	14 (61%)	0.736
Smoker (%)			0.372
Current	3 (25%)	2 (9%)	
Ex	2 (17%)	3 (13%)	
Body mass index (kg/m^2^)	50.4 ± 6.7	48.2 ± 7.8	0.424
Systolic BP (mmHg)	147 ± 12.3	145 ± 17.3	0.775
Diastolic BP (mmHg)	94 ± 10.9	94 ± 8.1	0.909
Glucose (mg/dL)	142 ± 60.0	140 ± 58.1	0.923
HbA1C (%)	7.4 ± 0.8	7.3 ± 0.8	0.852
Hyperuricemia (mg/dL)	5.8 ± 1.2	5.6 ± 1.4	0.688
Total Cholesterol (mg/dL)	195 ± 44.9	186 ± 44.5	0.571
LDL Cholesterol (mg/dL)	122 ± 29.5	117 ± 39.4	0.681
HDL Cholesterol (mg/dL)	39 ± 8.7	45 ± 9.1	0.084
Triglycerides (mg/dL) *	141 (1162; 214.7)	120 (102; 165)	0.161
Dyslipidaemia (%)	11 (92%)	14 (61%)	0.083
Type 2 diabetes mellitus (%)	9 (75%)	17 (74%)	0.944

Data are given as mean ± SD or percentage (%), except: * median (P25; P75).

## Data Availability

The data presented in this study are available on request from the corresponding author.

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
