# Peer review of "Arterial Hypertension in Morbid Obesity after Bariatric Surgery: Five Years of Follow-Up, a Before-And-After Study"

_ijerph, 2022, doi:10.3390/ijerph19031575_

Round 1

Reviewer 1 Report

The authors conducted a prospective observational study to analyze the alterations of blood pressure in 273 patients with morbid obesity and received gastric bypass surgery. The follow-up period lasted for 5 years and patients were divided into hyper-and normotensives. Several factors were identified related to hypertension remission and also of recurrence of hypertension. Several concerns are listed as below:

  1. The background information is lacking such as duration of hypertension, development of complications, pharmacological intervention.
  2. Almost 80% patients were women, it will be interesting to examine if there is gender difference in their study.
  3. How to exclude secondary hypertension? Is there any renal disease? Or endocrinological disorders?
  4. The logistic regression as shown in table 3 indicated diabetes was independent predictor; however, table 2 revealed more diabetes was associated with remission. But their interpretation described higher hyperuricemia in diabetics. This result is rather odd.
  5. Comparison with previous studies did not reveal novel finding and mechanism(s) explaining obesity-associated hypertension. The authors can provide other laboratory data or biomarkers to illustrate the alterations of hypertension in the present study, including remission and recurrence.

Author Response

Thank you very much for the comments and time spent reading this work.

The authors conducted a prospective observational study to analyze the alterations of blood pressure in 273 patients with morbid obesity and received gastric bypass surgery. The follow-up period lasted for 5 years and patients were divided into hyper-and normotensives. Several factors were identified related to hypertension remission and also of recurrence of hypertension. Several concerns are listed as below:

  1. The background information is lacking such as duration of hypertension, development of complications, pharmacological intervention.

The information corresponding to the pharmacological intervention has been included in Table 1, including polymedication and the number of comorbidities in the patients. Regarding the complications of the surgery, we have added in the Results section the fact that 17% of the patients (42) needed a second intervention. However, no significant differences were found (p-value = 0.398) between the group with hypertension at the beginning of the study, 19.3%, and the normotensive group at the beginning of the study, 14.8%).

  1. Almost 80% patients were women, it will be interesting to examine if there is gender difference in their study. 

We have performed an analysis of the difference by gender and it has been included in the Materials and Methods and Results sections, and the following sentence has been added: “There are no significant differences in sex (p = 0.744) and neither in the difference in evolution over time, (p = 0.617, interaction time * sex).

  1. How to exclude secondary hypertension? Is there any renal disease? Or endocrinological disorders?

Patients with kidney disease or those whose hypertension was of an endocrine origin, according to the protocol of the surgery service, were not included for surgical intervention.

  1. The logistic regression as shown in table 3 indicated diabetes was independent predictor; however, table 2 revealed more diabetes was associated with remission. But their interpretation described higher hyperuricemia in diabetics. This result is rather odd.

We believe that this was not well understood due to an incorrect wording of the sentence, since there was an error that has now been modified in the manuscript. “initial SBP and total cholesterol, as well as with higher hyperuricemia in diabetics” has now been replaced by “.... initial SBP and total cholesterol, as well as with higher hyperuricemia and diabetes”.

  1. Comparison with previous studies did not reveal novel findings or mechanisms explaining obesity-associated hypertension. The authors can provide other laboratory data or biomarkers to illustrate the alterations of hypertension in the present study, including remission and recurrence.

The determination of biomarkers has not been performed in the laboratory in the present study and we have now added this as a limitation of the work.

Reviewer 2 Report

The authors analyzed the prevalence and evolution of hypertension (HTN) and weight loss in patients suffering from morbid obesity before and after bariatric surgery (Laparoscopic Roux-En-Y Gastric Bypass), during a follow-up period of 5 years. Bariatric surgery in obese patients was associated with an improvement of HTN and no weight loss differences were observed between the group of patients showing HTN remission at 2 years and the group who did not.

  1. How was the improvement in hypertension determined? How severe should the decrease in SBP or DBP be?
  2. How was the blood pressure and its reduction monitored/measured after surgery? Was it measured the same way as at baseline?
  3. Was sleep apnoea excluded in your patients? Did you perform polysomnography?
  4. Have you ruled out secondary hypertension?
  5. Have you performed echocardiography on your patients? It would be interesting to have these data.
  6. Page 7, line 213

         “Authors In this study…” – please check and correct the typos

Author Response

Thank you very much for the comments and time spent reading this work.

The authors analyzed the prevalence and evolution of hypertension (HTN) and weight loss in patients suffering from morbid obesity before and after bariatric surgery (Laparoscopic Roux-En-Y Gastric Bypass), during a follow-up period of 5 years. Bariatric surgery in obese patients was associated with an improvement of HTN and no weight loss differences were observed between the group of patients showing HTN remission at 2 years and the group who did not.

  1. How was the improvement in hypertension determined? How severe should the decrease in SBP or DBP be?.

Thank you for the comment and we agree with you, we understood the term improvement to be equivalent to that of remission, when it is not. We have removed this terminology and introduced the concepts of remission and relapse/recurrence, as they now appear in the Materials and Methods section.

  1. How was the blood pressure and its reduction monitored/measured after surgery? Was it measured the same way as at baseline?.  

We recorded the patients’ hypertension data in the consultation prior to surgery and during the 5-year follow-up period. BP measurements were performed by the same group of researchers with standardization of the processes and technique, as described in the Materials and Methods section.

  1. Was sleep apnoea excluded in your patients? Did you perform polysomnography?

We have not excluded patients with sleep apnea. The total number of OSAS (apnea) patients of the 247 patients in the study is 62 (25.1%). No significant differences (p = 0.632) were found between the group with sleep apnea in hypertensive patients, 32 (26.9%), and normotensive patients, 30 (23.4%). We did not perform polysomnography on the patients, but they had been diagnosed prior to the study and we considered this as a comorbidity.

  1. Have you ruled out secondary hypertension?

Patients with kidney disease or those whose hypertension was of endocrine origin, according to the protocol of the surgery service, were not included for surgical intervention

Have you performed echocardiography on your patients? It would be interesting to have these data.

Electrocardiograms and echocardiograms were performed on all patients before bariatric surgery, ruling out the inclusion in bariatric surgery and therefore the inclusion of those with alterations in the study.

  1. Page 7, line 213

         “Authors In this study…” – please check and correct the tipos.

Thank you for identifying this tipo which has now been corrected in the text and the manuscript has been checked for tipos.

Reviewer 3 Report

Interestning data, especially the 5 year control. There are some concerns here:

drugs: how many subjects did take drugs at inclusion, what type of drug, and how many discontinued with drugs?

Please define "improvement as shown by 60%? 

Figures shows good efficacy on BMI, but not in table 2 and 4. Some error, as well with hypertension in these tables, is still there even in "without hypertension".

Discussion is bit too long, more a kind of review of this issue, much is well known for the interested reader.

Author Response

Thank you very much for the comments and time spent reading this work.

Interesting data, especially the 5 year control. There are some concerns here:

drugs: how many subjects did take drugs at inclusion, what type of drug, and how many discontinued with drugs?

The information corresponding to the pharmacological intervention has now been included in Table 1, including polymedication and the number of comorbidities in the patients.

Please define "improvement as shown by 60%? 

Thank you for this comment and we agree with you. The word improvement was not very well applied.  In the new version of the manuscript we now refer, at all times, to remission and relapse/recurrence and as such this has now been modified in the text.

Figures shows good efficacy on BMI, but not in table 2 and 4. Some error, as well with hypertension in these tables, is still there even in "without hypertension".

Figures 1 show the evolution of weight loss in normotensive and hypertensive patients at the beginning of the study, and Figure 2 shows this evolution with respect to remission of hypertension or not at 5 years.

Tables 2 and 4 show the BMI values of the patients at baseline who remit at 2 years or recur at 5 years in the study period, respectively. In both cases, it shows that the initial BMI was similar in the patients who remit or recur.

Discussion is bit too long, more a kind of review of this issue, much is well known for the interested reader.

We have now tried to shorten some paragraphs in the Discussion section.